
# Interpolation uncertainty of atmospheric temperature radiosoundings

Alessandro Fassó[1], Michael Sommer[2], and Christoph von Rohden[2]

[1]University of Bergamo, Italy
[2]GRUAN Lead Centre, Deutscher Wetterdienst, Lindenberg, Germany

**Correspondence:** Alessandro Fassó (alessandro.fasso@unibg.it)

**Abstract.** This paper is motivated by the fact that, although temperature readings made by Vaisala RS41 radiosondes at GRUAN sites (www.gruan.org) are given at $1\,\mathrm{s}$ resolution, for various reasons, missing data are spread along the atmospheric profile. Such a problem is quite common in radiosonde data and other profile data. Hence, (linear) interpolation is often used to fill the gaps in published data products. In this perspective, the present paper considers interpolation uncertainty. To do this,

a statistical approach is introduced giving some understanding of the consequences of substituting missing data by interpolated ones.

In particular, a general frame for the computation of interpolation uncertainty based on a Gaussian process (GP) set-up is developed. Using the GP characteristics, a simple formula for computing the linear interpolation standard error is given. Moreover, the GP interpolation is proposed as it provides an alternative interpolation method with its standard error.

For the Vaisala RS41, the two approaches are shown to give similar interpolation performances using an extensive cross-validation approach based on the block-bootstrap technique. Statistical results about interpolation uncertainties at various GRUAN sites and for various missing gap lengths are provided. Since both provide an underestimation of the cross-validation interpolation uncertainty, a bootstrap-based correction formula is proposed.

Using the root mean square error, it is found that, for short gaps, with an average length of $5\,\mathrm{s}$, the average uncertainty is
smaller than $0.10\,\mathrm{K}$. For larger gaps, it increases up to $0.35\,\mathrm{K}$ for an average gap length of $30\,\mathrm{s}$, and up to $0.58\,\mathrm{K}$ for a gap of $60\,\mathrm{s}$.

## 1   Introduction

The quality of climate variable profiles in the upper troposphere - lower stratosphere (UTLS) is relevant in various scientific fields. In particular, it is important for numerical weather prediction, satellite observation validation and climate change
understanding, including extreme events such as droughts and tornadoes.

The GCOS (Global Climate Observing System) Reference Upper-Air Network (GRUAN, www.gruan.org) is a network for reference measurements of UTLS (Seidel et al., 2009; Bodeker et al., 2016). Immler et al. (2010) discussed the concepts of reference measurements, traceability, full metadata description, a proper manufacturer-independent instrument characterization, and the assessment of measurement uncertainties for upper-air observations.



In this frame, GRUAN data processing for the Vaisala RS92 radiosonde was developed to meet the above criteria for reference measurements (Dirksen et al., 2014). The related data product is characterised not only by the above mentioned metrological requirements but also by high-vertical-resolution. After the introduction of the new Vaisala RS41 radiosonde, GRUAN is currently developing the corresponding data processing for the new instrument (Dirksen et al., 2019).

Although temperature readings made by Vaisala RS41 radiosonde at GRUAN stations are given at $1\,\mathrm{s}$ resolution, for various reasons, missing data are sometimes present along the atmospheric profile. If one is led to interpolate the missing measurements, since an interpolation error is implied, the related uncertainty is to be considered in the uncertainty budget.

The interpolation of atmospheric profiles has been considered in the literature from various points of view. In some cases, interpolation is applied to the measurement uncertainty. For example, considering the AERONET Version 3 aerosol retrievals, Sinyuk et al. (2020) obtain the uncertainty by interpolation of a look up table.

A second and more relevant use of interpolation is related to the measurement itself. In this field, Ceccherini et al. (2018) used interpolation for data fusion of Ozone satellite vertical profiles. Interpolation uncertainty and more generally co-location uncertainty has been computed using simulated profiles. Similarly, in co-location uncertainty of total ozone, Verhoelst et al. (2015) contemplate interpolation in the so-called OSSSMOSE simulator.

In the frame of radiosonde co-location uncertainty, considering relative humidity, Fassó et al. (2014) used a statistical approach based on the heteroskedastic functional regression model. Considering pressure, Ignaccolo et al. (2015) extended the latter approach to a 3D functional regression approach. In these two papers, the interpolation uncertainty is implicitly assessed by means of the model error variance.

The comparisons of radiosonde and satellite data are often based on low-vertical-resolution radiosonde profiles measurements such as the data collected within the network of the Universal Rawinsonde Observation Program (RAOB) because of their global coverage. In some cases interpolation is not required because of the higher vertical resolution of satellite profiles (Sun et al., 2010). In other cases, interpolation is required. For example, Finazzi et al. (2019) considered the harmonisation of the low-vertical-resolution RAOB temperature and humidity radiosonde measurements and the corresponding atmospheric profiles derived from the Infrared Atmospheric Sounding Interferometer (IASI) aboard Metop-A and Metop-B satellites. In this frame spline interpolation of RAOB profiles was indirectly assessed through a comparison with GRUAN radiosonde reference measurements.

As a common trait of the above literature, interpolation of atmospheric profiles is quite common, but a systematic analysis of interpolation uncertainty per se is not yet available. A general approach to interpolation is the Geostatistics approach (Cressie and Wikle, 2011) which is the same as the Gaussian Process (GP) approach (Rasmussen and Williams, 2006) to a large extent. Its value is due to the fact that it gives optimal interpolation under some conditions. With some variations, the related optimal interpolation algorithm is based on the autocovariance function characterising also the structure function (Sofieva et al., 2008). This approach is often used to interpolate in a higher dimensional space such as the Euclidean plane, the sphere (Alegria et al., 2017), the three-dimensional Euclidean space or the circular shell representing the atmosphere. Interpolating and forecasting is sometimes overlapping, in particular this happens when the GP is defined, for example, on time cross a sphere (Porcu et al., 2016). Interestingly, it can be shown that the spline interpolation is a special case of the GP interpolation (Kimeldorf and



Wahba, 1970). Another interesting point is that the GP approach comes with a formula for interpolation uncertainty estimation. It must be noted that the formula is correctly used if the "true data generation mechanism" is a GP. If the GP is simply "an approximation" an additional term must be added.

In this paper, the uncertainty of the one-dimensional linear interpolation is discussed integrating two approaches. In the first stage, the closed form formula of the linear interpolation uncertainty is presented under the assumption that the observed

atmospheric profile is generated by a GP. In the second phase, thanks to the availability of appropriate data, the GP assumption is relaxed and a block-bootstrap estimator is constructed. The approach is valid for any atmospheric profile dataset. Considering the motivating application, which focuses on temperature readings of Vaisala RS41 at GRUAN sites, the objective of this paper is to contribute to the understanding of interpolation uncertainty expressed as a function of missing gap length, missing frequency, altitude and site.

To do this, "good" launches without missing data are used. Each profile is divided in a learning set and a testing set, the latter being used as missing data for interpolation uncertainty assessment. This is done for various missing patterns that resemble observed "bad" launches, which are characterised by many missing measurements. In particular, increasing gap average lengths will be analysed. The testing sets will be extracted using a block-bootstrap cross-validation scheme. Hence although the numerical results are specific to Vaisala RS41 temperature data set, the approach is quite general and may be

applied to other sensors.

The rest of the paper is organised as follows. Section 2 motivates the paper by discussing the soruces of gaps in data reception and their impact in GRUAN data processing. Section 3 introduces the Gaussian Process (GP) set up used to provide the formal assessment of linear interpolation uncertainty and to introduce the GP interpolation with its standard deviation. Section 4 presents the data sets, which are related to Vaisala RS41 observations at seven GRUAN sites and are used in the empirical

analysis. Section 5 describes the re-sampling technique able to simulate random patterns of missing values of different duration. Section 6 describes the cross-validation scheme essential for the uncertainty computations and the model selection, which is discussed in Section 7. Section 8 presents the results, compares the behaviour of the two interpolation techniques and proposes an empirically corrected formula for interpolation uncertainty. Eventually, Section 9 draws some conclusions.

## 2  Data processing and interpolation

There are several possible reasons for temporary gaps in data reception. These include the presence of obstacles that may interfere with radio transmission to the ground station (trees, buildings, local geography), extraordinary meteorological conditions, or instrument-related reasons. Considering an ascent as a trajectory rather than a vertical profile, it seems evident that the probability for the occurrence of data gaps tends to increase with the horizontal distance from the launch site (weaker radio signal), which can significantly exceed the vertical distance depending on wind conditions. The GRUAN Lead Centre conducted a sta-

tistical analysis for the occurrence of data gaps in RS41 radiosonde soundings performed at 15 GRUAN stations in the period 2014-2019. The results show that gaps occur in more than 20% of the soundings, virtually independent of the height ranges, with the majority (>95%) having less than 15 gaps per 1000 s (=1000 data points). Up to 30 km, gaps >10 s only play a role

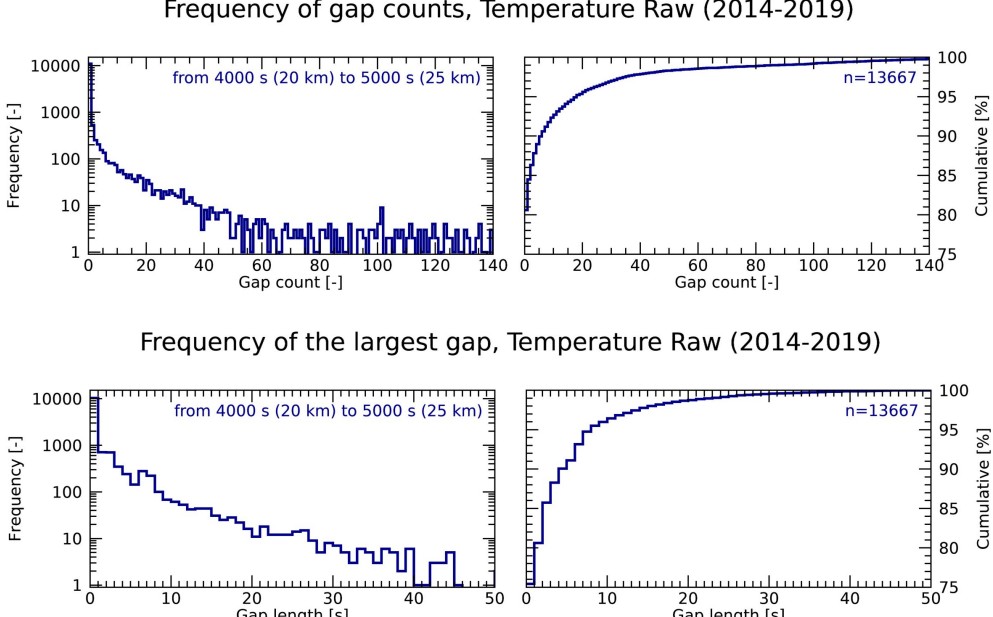

**Figure 1.** Top left: Frequency distribution of the number of data gaps (independent of gap length). Bottom left: Frequency distribution of the length of the largest gap identified in a sounding. The two right panels show the corresponding cumulative frequencies.

in about 5% of the ascents, however the occurrence of larger gaps generally increases with height (distance). Figure 1 gives an example for the stratospheric height section between 20 km and 25 km, where 13'667 profiles are included.

The GRUAN data processing is based on the raw data from the physical radiosonde's sensors, namely temperature, relative humidity, positioning data (GNSS), and also pressure if an on-board sensor is present. Corrections to the raw data for known or experimentally evidenced systematic effects are applied. For example adjustments from pre-flight ground checks, corrections of sensor time lags, or solar radiative effects. Some intermediate variables are in turn calculated (e.g., effective air speed or ventilation) as components of the correction algorithms. A number of secondary variables are finally derived, for example

altitude, geopotential height, water vapour content, or wind components. At different processing stages, smoothing filters are applied for estimation and separation of noise components of the signal. Through all these steps the regular grid of the measured raw data is maintained, that is, all variables and uncertainties in the product variables are given with the original high resolution.

This procedure inevitably leads to certain technical difficulties if data gaps randomly or intermittently occur. For example, smoothing with certain filter kernel lengths easily may introduce effects which are difficult to handle when running over gaps on

the regular grid or - even more - when running into larger gaps comparable to or exceeding the actual kernel length. The same applies to uncertainty estimates to be associated with the averaged (smoothed) values. Another example is related to magnitudes which are calculated cumulatively with height, such as pressure derived from positioning, temperature, and humidity data, or the integrated water vapour content. As a consequence there may be processing-related irregularities or deviations in the profile





data and uncertainty estimates, the systematics and extent of which is difficult to predict. Depending on the purpose for which
the GRUAN data product are further used (e.g., process studies based on high-resolution data, or average-based long-term
studies for climate) such systematics may have different impact.

## 3 Interpolation uncertainty

In this section, formulas of the uncertainty for both linear and stochastic interpolation are considered under some stochastic
assumptions about the data generation mechanism.

In particular, considering a radiosonde flight, we assume that $t = 1, ..., T$ is the flying time in seconds from take off and $y(t)$
is the observed temperature in Kelvin given by the following measurement error equation

$$y(t) = s(t) + \epsilon(t). \tag{1}$$

In model (1), $s(t)$ is the unobserved "true" temperature with a local dynamics described by a Gaussian Process (GP) char-
acterised by a power exponential autocovariance function (Cressie and Wikle, 2011; Rasmussen and Williams, 2006). Hence,
conditionally on some unobserved time-dependent atmospheric conditions denoted by $a(t)$, the GP $y(t)$ has the following
autocovariance function:

$$\gamma(t - t', a(t)) = \sigma_s^2 \exp(-|t - t'|^p / \theta^p) + \sigma_\epsilon^2 I(t = t') \tag{2}$$

where $p = 1, 2$, the dependence on $a(t)$ is omitted in the right hand side for notational simplicity, and $I$ is the indicator function,
that is $I = 1$ if $t = t'$ and zero else.
In (2), the variance of $y(t)$ is given by

$$\sigma_y^2 = \sigma_s^2 + \sigma_\epsilon^2 \tag{3}$$

where $\sigma_s > 0$ is the standard error of $s(t)$, and $\sigma_\epsilon \geq 0$ is the measurement uncertainty, $\sigma_\epsilon^2 = E(\epsilon^2)$. For the instruments installed
on the Vaisala RS41 it is known that the sensor-intrinsic "noise" of a temperature sensor is very small ($< 0.01$ K), hence we
expect to find a small $\sigma_\epsilon$ for the data of this paper. In addition, $\theta > 0$ represents the atmospheric persistence range.
The GP is characterised by the parameter set $\boldsymbol{\Psi} = (\theta, \sigma_s, \sigma_\epsilon)$, which is assumed to be slowly varying in time, hence charac-
terising locally the atmospheric conditions $a(t)$:

$$\boldsymbol{\Psi} = \boldsymbol{\Psi}_{a(t)}. \tag{4}$$

Note that, from the practical point of view, the random error $\epsilon$ is a Gaussian white noise and $\sigma_\epsilon$ represents the random un-
certainty, while (vertically) correlated errors could be confused with $s(t)$. This point will be considered further in Section
8.





### 3.1 Linear interpolation

Considering an observation gap in the interval $(t^-, t^+)$, the linear interpolator at time $t$, for $t^- \leq t \leq t^+$, is straightforwardly defined by the following formula

$$m(t) = (1 - \alpha(t))y^- + \alpha(t)y^+ \tag{5}$$

where, $y^{\pm} = y(t^{\pm})$, and $\alpha(t) = \frac{t - t^-}{t^+ - t^-}$.

In general, the squared interpolation uncertainty

$$u(t)^2 = E[(m(t) - s(t))^2] \tag{6}$$

is defined in terms of the true signal $s(t)$ and may be related to the interpolation Mean Square Error

$$MSE_y(t)^2 = E[(m(t) - y(t))^2]$$

by the well known relation $u(t)^2 = MSE_y(t)^2 + \sigma_\epsilon^2$.

Since, using field observations, only $MSE_y(t)$ may be directly estimated, if the measurement uncertainty $\sigma_\epsilon$ is unknown, estimating $u(t)$ may be an issue, and a statistical modelling approach is important.

Assuming that the true signal $s$ is a GP as above discussed, the Appendix shows that the linear interpolation uncertainty given in Equation (6) may be computed by the following Standard Error (SE) formula:

$$
\begin{aligned}
\quad SE(t)^2 \;=\; & 2\sigma_y^2 \left\{ 1 - \alpha + \alpha^2 \right\} \\
& + 2 \left\{ \alpha(1 - \alpha)\gamma(t^+ - t^-) - \alpha\gamma(t^+ - t) - (1 - \alpha)\gamma(t - t^-) \right\} \\
& + \sigma_\epsilon^2
\end{aligned} \tag{7}
$$

where, with abuse of notation, $\alpha = \alpha(t)$. Note that, $SE(t)^2 = u(t)^2$ if the GP assumption is satisfied, but two different symbols are used because in Section 8 this assumption will be relaxed.

Equation (7) defines a function of $t$ which depends on the atmospheric persistence modelled by $\gamma$ and the gap size $t^+ - t^-$. Since $\gamma$ is not continuous in zero, the same happens to $SE(t)$ at the gap interval borders.

Figure 2 considers the case where $s(t)$ is a white noise, that is $\gamma(h) = 0$ for $h \neq 0$ and $\gamma(0) = \sigma_y^2$. At the gap borders, the interpolation is error free, $m(t^{\pm}) = y^{\pm}$, and the uncertainty is $u(t^{\pm}) = \sigma_\epsilon$. For $t$ strictly inside the gap interval, we have

$$\frac{3}{2}\sigma_y^2 + \sigma_\epsilon^2 \;\leq\; u(t)^2 \;<\; 2\sigma_y^2 + \sigma_\epsilon^2$$

where the minimum is achieved in the center of the gap interval. In this particular case, the uncertainty range does not depend on the gap size.

The above thresholds may be overcome in the presence of correlation. In general for a GP with $\theta > 0$ the uncertainty depends both on the GP characteristics and the gap size. As an illustration, using $\sigma_s = 0.5K$, $\sigma_\epsilon = 0.01K$, and $\theta = 3\,\mathrm{s}$, Figure 3 shows how the interpolation uncertainty depends on the gap size and on the distance from the observations in presence of short 165 correlation range. More interestingly for applications, Figure 4 shows that the linear interpolation uncertainty strongly depends on the correlation range.



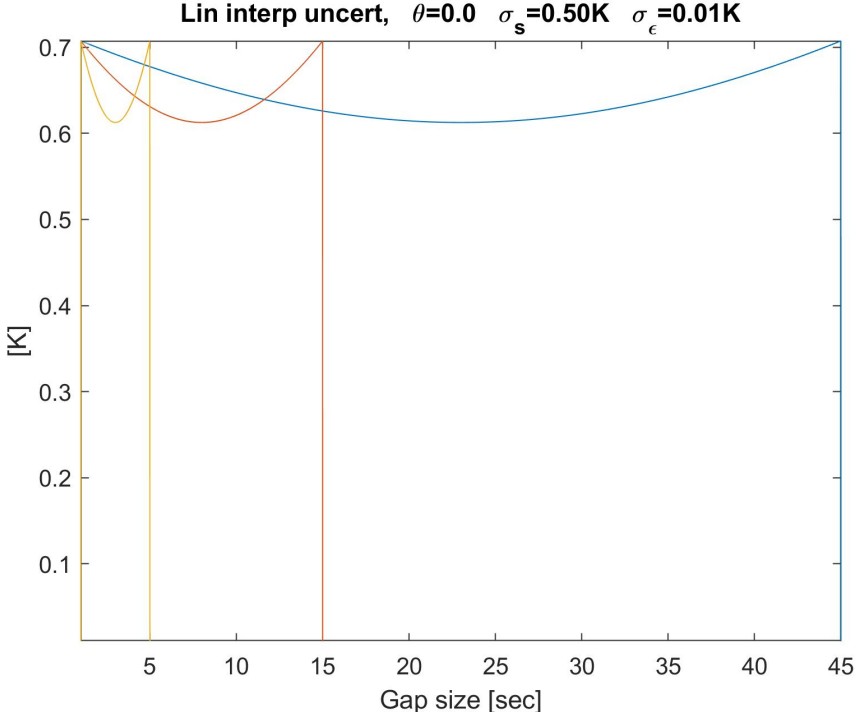

**Figure 2.** Linear interpolation SE, Equation (7), as a function of the distance from observations for a white noise process with $\sigma_s = 0.5\,\mathrm{K}$ and $\sigma_\epsilon = 0.01\,\mathrm{K}$. Three gap sizes are considered (45, 15 and 5 s) .

### 3.2 Gaussian Process interpolation

The assumption that the temperature profile $y(t)$ is a realisation of a GP may be extended to cover for a non constant mean so that, using some basis functions $\boldsymbol{h}()$, model (1) is rewritten as

$y(t) = \boldsymbol{h}(t)'\beta + s(t) + \epsilon(t)$

with parameter set $\boldsymbol{\Psi} = (\beta, \theta, \sigma_\epsilon, \sigma_s)$. In this context, Equation (3) defines the variance of $y(t)$ conditional on $\boldsymbol{h}(t)'\beta$, namely $Var(y(t)|\boldsymbol{h}(t)'\beta)$. Let us denote the set of all non missing observations during the radiosonde flight by $\boldsymbol{Y}$, the matrix of the corresponding basis functions by $\boldsymbol{H}$, and assume that $\boldsymbol{\Psi}$ is known. Then the GP interpolation of a missing observation at time $t^*$ is given by the well known conditional expectation formula

$m(t^*) = E(y(t^*)|\boldsymbol{Y}) = \boldsymbol{h}(t^*)'\beta + \boldsymbol{\Sigma}'_{y(t^*),\boldsymbol{Y}}\boldsymbol{\Sigma}^{-1}_{\boldsymbol{Y},\boldsymbol{Y}}(\boldsymbol{Y} - \boldsymbol{H}\beta)$   (8)

where $\boldsymbol{\Sigma}_{\boldsymbol{Y},\boldsymbol{Y}}$ is the covariance matrix of the good observations $\boldsymbol{Y}|\boldsymbol{H}\beta$, and $\boldsymbol{\Sigma}_{y(t^*),\boldsymbol{Y}}$ is the covariance vector between the missing observation $y(t^*)|\boldsymbol{h}(t^+)'\beta$ and $\boldsymbol{Y}|\boldsymbol{H}\beta$. In addition to point estimation, the GP approach provides also the interpolation

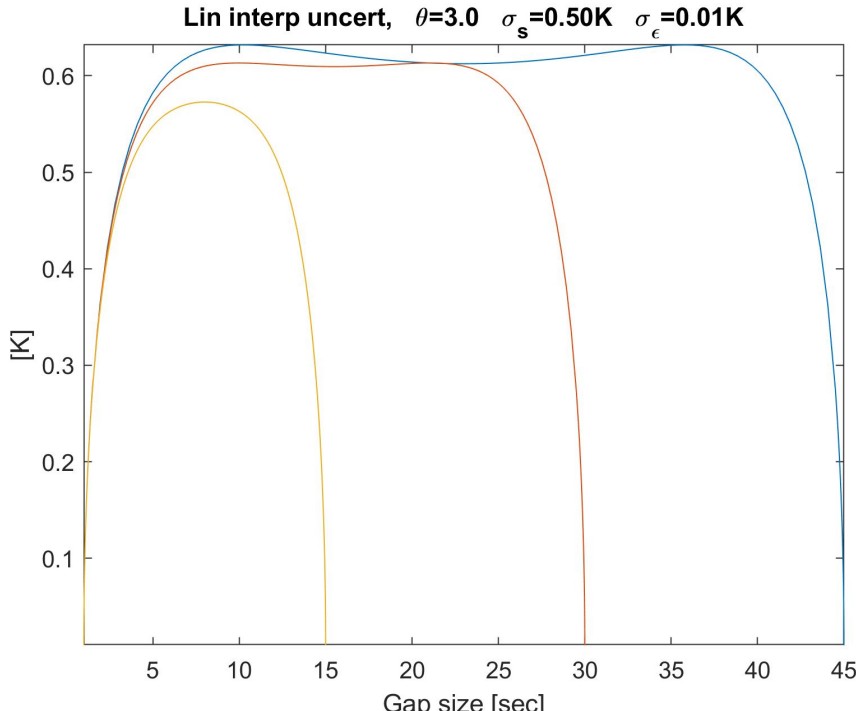

**Figure 3.** Linear interpolation SE, Equation (7), as a function of the distance from observations for a GP with $\sigma_s = 0.5\,\mathrm{K}$, $\sigma_\epsilon = 0.01\,\mathrm{K}$, and $\theta = 3\,\mathrm{s}$. Three gap sizes are considered (45, 30 and 15 s) .

standard error:

$$SE(t^*)^2 = E(m-y)^2 = \sigma_y^2 - \mathbf{\Sigma}'_{y(t^*),Y} \mathbf{\Sigma}^{-1}_{Y,Y} \mathbf{\Sigma}_{y(t^*),Y} \tag{9}$$

which can be used as an estimate of the interpolation uncertainty, provided the GP is a good description of the problem under study and $\mathbf{\Psi}$ is approximately known.

## 4 Data

Two datasets provided by the GRUAN Lead Centre (www.gruan.org/network/lead-centre), and related to the seven GRUAN stations of Table 1, are considered here. One is named $Few\_nan$ in this paper and contains 276 temperature profiles charac-
185 terised by "little" missing data. The second one, named $Many\_nan$, contains 273 profiles with "many" missing.

As a preliminary analysis of the "bad" dataset $Many\_nan$, Figure 5 gives the distribution of the fraction of missing data per launch. The average missing fraction is 0.13, and the average gap length is 3.6 s. These values will be used to set the parameters of the simulated gap patterns of Section 5.



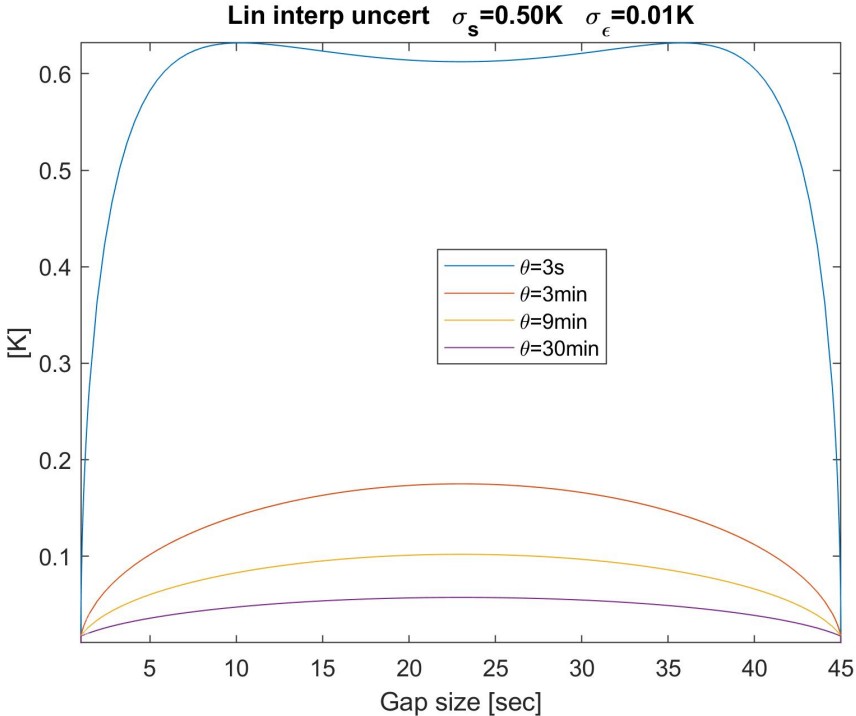

**Figure 4.** Linear interpolation SE, Equation (7), as a function of the distance from observations for a GP's with $\sigma_s = 0.5\,\mathrm{K}$, $\sigma_\epsilon = 0.01\,\mathrm{K}$, and $\theta = 3\,\mathrm{s}, 3\,\mathrm{min}, 9\,\mathrm{min}$ and $30\,\mathrm{min}$.

For further interpolation analysis, those profiles in $Few\_nan$ with very little missing gaps are selected. In particular, the

L=177 launches which have gaps shorter than 5 seconds, and a total of less then 10 missing values per profile have been used in this paper. The profile duration distribution is depicted in Figure 6, with an average profile duration of about 6000 seconds. This gives a total of more than one million measurements, which will be amplified using the bootstrap technique of Section 5.

## 5   Block-bootstrap cross-validation scheme

The block-bootstrap is a well-known technique (Politis and Romano, 1994; Mudelsee, 2014) for generating synthetic time se-

ries replicates, and, in this paper, is used to construct the cross-validation scheme. Let us consider a fully observed temperature profile, without missing values and, hence, measurements $y$ taken every second from take off, $t = 1, ..., T$: $\boldsymbol{Y} = (y(1), ..., y(T))$. This section presents a rule for partitioning each original profile $\boldsymbol{Y}$ as follows

$$\boldsymbol{Y} \longrightarrow [\boldsymbol{Y}^L, \boldsymbol{Y}^*] \tag{10}$$





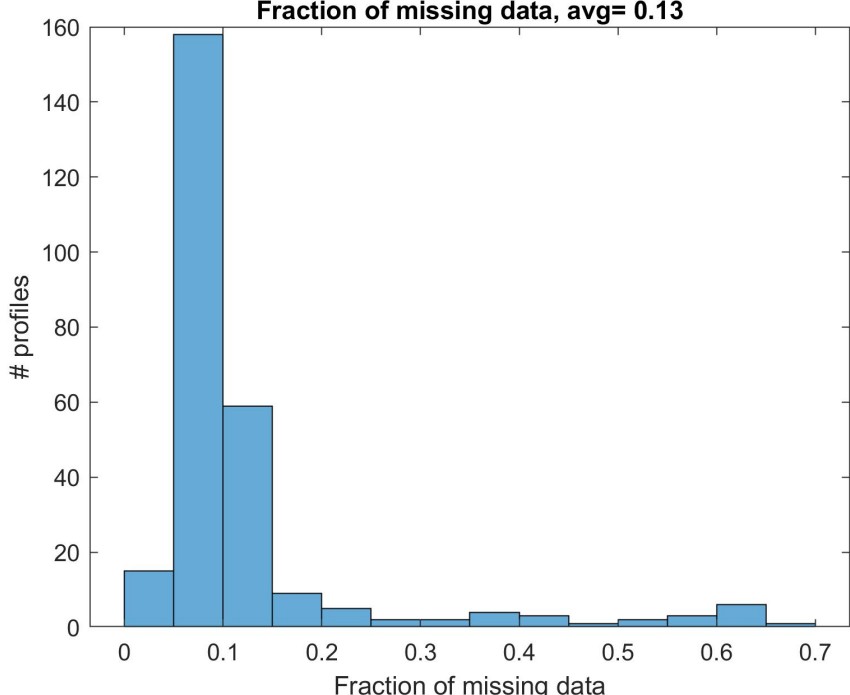

**Figure 5.** Frequency distribution of missing data fraction in $Many\_nan$ dataset.

| Station | Code | Country | Imported | Selected |
|---------|------|---------|----------|----------|
| Beltsville | BEL | USA | 33 | 15 |
| Lauder | LAU | NZ | 32 | 32 |
| Lindenberg | LIN | DE | 54 | 45 |
| Ny-Alesund | NYA | DE/FR | 35 | 35 |
| Payern | PAY | CH | 100 | 30 |
| Lamont | SGP | USA | 18 | 16 |
| Sodankylä | SOD | FI | 4 | 4 |
| | | | 276 | 177 |

**Table 1.** GRUAN stations and launch numbers from $Few\_nan$ dataset.



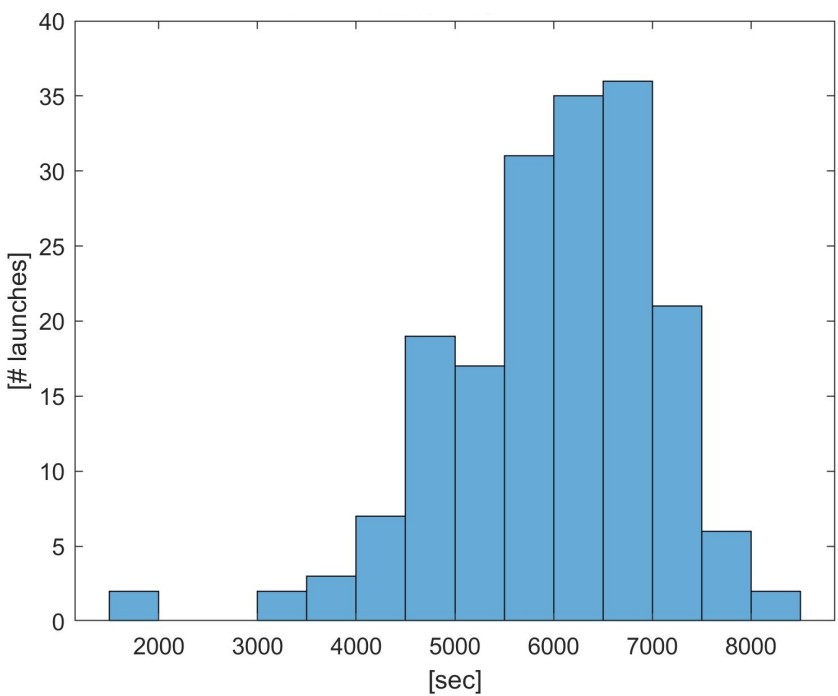

**Figure 6.** Frequency distribution of profile duration in $Few\_nan$ dataset.

where $\boldsymbol{Y}^L$ is the learning set – used for fitting – and $\boldsymbol{Y}^*$ is the validation set – used for testing and bootstrap-correction. In order to construct the testing set, $n_G$ gap sequences of average duration $\mu_G[\mathrm{s}]$ are extracted from the temperature profile $\boldsymbol{Y}$ and moved to the testing set $\boldsymbol{Y}^*$. Observe that, if the testing size (average) fraction is denoted by $f$, then $n_G = T \times f / \mu_G$.

The gap scheme is obtained by randomly generating and sorting the $n_G$ gap starting points $1 \leq t_1^* \leq ... \leq t_{n_G}^* \leq T$ and by building, for each of them, a gap sequence

$$t_j^*, ..., t_j^* + g_j$$

where the gap duration $g_j$ is a $Geometric$ random variable with mean $\mu_G$. In particular the length $g_j$ is truncated at $t_{j+1}^* - t_j^* - 1$ to avoid overlapping among different gap sequences. Let the resulting testing set index be denoted by $\mathbf{t}^*$. Ignoring above truncation, $\mathbf{t}^*$ has random dimension $n^* = n_G + \sum_{j=1}^{n_G} g_j$ and expected dimension $E(n^*) = T \times f$. Hence, the partitioning rule in (10) is defined by the testing set $\boldsymbol{Y}^* = (y(t), t \in \mathbf{t}^*)$ and the learning set $\boldsymbol{Y}^L = (y(t), 1 \leq t \leq T, t \notin \mathbf{t}^*)$.

We are interested in collecting information about the interpolation error in a dense vertical grid, even if the testing fraction $f$ is small. To do this in the application developed below, the above random extraction process is repeated $B$ times. So that for each observed profile $\boldsymbol{Y}$, $B$ replications are generated, namely

$$[\boldsymbol{Y}_b^L, \boldsymbol{Y}_b^*], \; b = 1, ..., B.$$





These replications give a statistical basis to assess the interpolation uncertainty at all altitudes also for those stations with a limited number of available profiles.

## 6 Cross-validation

The main results of the next section are obtained using linear interpolation of temperature vs time, based on the neighbouring values, and GP interpolation given by the expectation of $\boldsymbol{Y}^*$ conditionally on $\boldsymbol{Y}^L$. As in the previous section, let us denote temperature, in Kelvin, by $y$ and flying time, in seconds, by $t = 1, ..., T$. The total flying time $T$ depends on the single profile and station but suffixes are not used here for notational simplicity. For each station $s = 1, ..., S$ and launch $l = 1, ..., L_s$, we have the interpolated values

$$\hat{y}(t^*|s,l) = m_j(t^*|s,l)$$

where $j = 1, 2$ denotes the linear or the GP interpolation respectively.

Each bootstrap replicate $[\boldsymbol{Y}_b^L, \boldsymbol{Y}_b^*]$, $b = 1, ..., B$ is used first to estimate the GP model coefficients $\boldsymbol{\Psi}$ by the maximum likelihood method as explained in the next section and denoted by $\hat{\boldsymbol{\Psi}}$. Then, the interpolated values $\hat{y}(t^*) = m_2(t^*|\hat{\boldsymbol{\Psi}})$ are computed for the simulated missing times $t^*$ in the test data set, $\boldsymbol{Y}_b^*$, and the cross-validation interpolation errors are computed as follows:

$$e = e(t^*|s,l,b) = \hat{y}(t^*|s,l,b) - y(t^*|s,l).$$

As a result, the interpolation MSE and the corresponding Root MSE (RMSE) are classified by station, altitude and gap length:

$$MSE(ALT, s, \mu_G) = avg(e^2|ALT, s, \mu_G) \tag{11}$$

where $ALT$ is the atmospheric output layering with 1km resolution and $avg(\cdot|s, ALT)$ is the average of all the cross-validation terms with $alt \in ALT$, launched from station $s$ and generated using gap size $\mu_G$.

## 7 Modelling details

The GP interpolator depends on the local structure $m_2(t) = m_2(t|\boldsymbol{\Psi}_{a(t)})$, where $\boldsymbol{\Psi}_{a(t)}$ is as in Equation (4). In order to make teh local GP modelling feasible and computationally efficient a block-partitioning structure has been assumed. This amounts at dividing the atmosphere in layers, which may be different from the output layers of the previous section. Each atmospheric layer identifies a block and the variance-covariance matrix for the entire profile is assumed block diagonal with a constant parameter set $\boldsymbol{\Psi}_a$ in each layer block. This is a special case of the spatial partitioning approach (Heaton et al. , 2019), but continuity at the layer borders is ignored here because borders have been deliberately located far from the gaps.

The GP model selection considered the two autocovariance functions $\gamma$ in Equation (2), various basis functions $\boldsymbol{h}()$, and various layering's of the atmosphere to define the appropriate concept of local model $\boldsymbol{\Psi}_a(t)$ of Equation (4). For each layer $a$,





local estimation has been performed using the maximum likelihood method. The above alternatives have been optimised using the RMSE applied to the block-bootstrap replicates of Section 5.

Considering the layering problem, the results where little sensitive to layer size variations and a 400" layer size has been used, as it provides both a reasonable computing time and a satisfactory atmospheric adaptation. The exponential autocovariance

function with $p = 1$ resulted in a smaller cross-validation RMSE comparing to the square exponential one ($p = 2$).

The best results for the basis functions have been obtained with a piecewise linear function of time. In this regard, also other predictor set-up have been considered: a piecewise quadratic function of time and vector predictor set-ups including altitude, coordinates and wind. Using these more complex models did not result in relevant improvement of RMSE or, worst, it resulted in problems of singularity of the information matrix at various combinations of stations and layers. Hence, invoking Okham's

razor and looking for a robust and general model set-up, we concluded for the simplest piecewise linear function of time.

## 8    Results

The bootstrap champaign of this section is aimed at assessing the uncertainty of the linear interpolation, Equations (5) and (7), and of the GP interpolation, Equations (8) and (9). The cross-validation design is based on a $3 \times 3$ combination of gap sizes $\mu_G$ and missing fractions $f$, centred on the characteristics of the $Many\_nan$ dataset. In detail, we use $\mu_G = 4, 10, 30, 60$

255    sec and $f = 0.05, 0.13, 0.20$. Moreover, in order to have uncertainty estimates with a high vertical density, the 2-fold block-bootstrap validation of Section 5 is replicated $B = 50$ times giving a data set with more than 51 million measurements for each combination of $\mu_G$ and $f$.

Figure 7 depicts the overall linear interpolation uncertainty at each GRUAN station for the above $3 \times 3$ simulation design using the RMSE. The clustered pattern of the nine curves clearly shows that the missing fraction $f$ has a minor influence on

the uncertainty in the range $0.05 - 0.20$, which is the range of interest for meaningful practical applications. Hence for the rest of the paper, we consider only the $Many\_nan$ dataset missing fraction, $f = 0.13$.

Table 2 summarises the RMSE of both the linear and GP interpolation. Overall, the average interpolation uncertainty is smaller than 0.1K for little gaps ($\mu_G = 4$"), increases to about 0.16 K for medium gaps ($\mu_G = 10$"), and increases further to 0.35 K and 0.58 K for large and very large gaps ($\mu_G =$30" and 60") respectively. Considering jointly Figure 7 and the latter

table, it can be observed that Lamont, Payerne and Lauder have slightly larger values at all gap sizes. Moreover, Table 2 shows that the two interpolation approaches have a very close RMSE. In fact, not only they have close performances, but, for any practical purpose, they are also exchangeable, since the mean absolute difference between the two is smaller 0.01 K. Hence in the rest of the paper, we do not replicate figures and results for both interpolation methods.

Figure 8 depicts the vertical behaviour of interpolation uncertainty at GRUAN stations, with average gap size increasing

from panel a) to panel c). As expected the uncertainty has a minimum near the tropopause. Moreover, after a fast increase, it stabilizes at a value often larger than the lower atmosphere uncertainty level. It is worth observing that the various stations have globally similar values, but again Lamont, Payern and Lauder have often the largest values.





| Station | Profiles | $\mu_G = 4''$ | | $\mu_G = 10''$ | | $\mu_G = 30''$ | | $\mu_G = 60''$ | |
|---|---|---|---|---|---|---|---|---|---|
| | | GP | Linear | GP | Linear | GP | Linear | GP | Linear |
| BEL | 15 | 0.084 | 0.088 | 0.159 | 0.160 | 0.338 | 0.363 | 0.590 | 0.604 |
| LAU | 32 | 0.106 | 0.107 | 0.180 | 0.184 | 0.370 | 0.389 | 0.599 | 0.612 |
| LIN | 45 | 0.073 | 0.074 | 0.145 | 0.145 | 0.314 | 0.324 | 0.548 | 0.542 |
| NYA | 35 | 0.072 | 0.073 | 0.127 | 0.130 | 0.269 | 0.269 | 0.463 | 0.460 |
| PAY | 30 | 0.098 | 0.098 | 0.180 | 0.181 | 0.370 | 0.391 | 0.659 | 0.658 |
| SGP | 16 | 0.107 | 0.109 | 0.189 | 0.187 | 0.401 | 0.420 | 0.703 | 0.698 |
| SOD | 4 | 0.074 | 0.076 | 0.137 | 0.138 | 0.281 | 0.363 | 0.426 | 0.478 |
| | 177 | 0.087 | 0.088 | 0.159 | 0.160 | 0.334 | 0.349 | 0.574 | 0.576 |

**Table 2.** Comparison of cross-validation RMSE between GP and linear interpolation for increasing average gap length $\mu_G = 4, 10, 30$ and $60\,\mathrm{s}$. Cross-validation is based on $B = 50$ block-bootstrap replications, each with missing fraction $f = 0.13$.

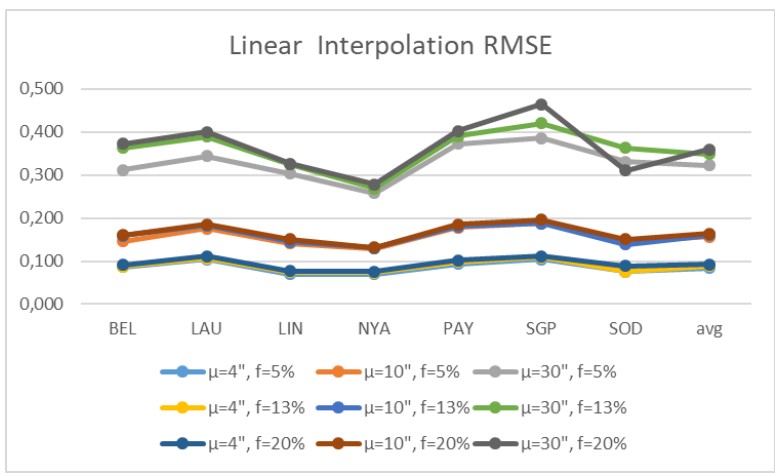

**Figure 7.** Linear interpolation uncertainty by GRUAN station and average gap size $\mu_G = 4, 10$ and $30\,\mathrm{s}$. The cross-validation uncertainty ($y$-axis) is based on the Root Mean Square Error (RMSE), for missing fractions $f = 0.05, 0.13$ and $0.20$.



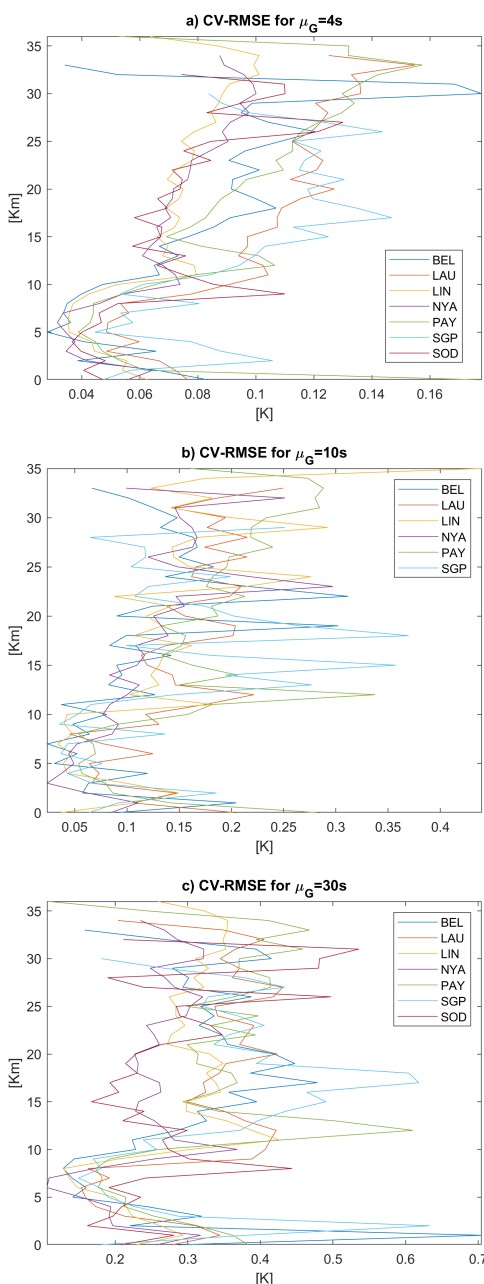

**Figure 8.** Linear interpolation uncertainty of GRUAN stations. The cross-validation uncertainty ($x$-axis) is based on the Root Mean Square Error (RMSE) and missing fraction $f = 0.13$. Panel a: average gap size is $\mu_G = 4\,\text{s}$; panel b: average gap size is $\mu_G = 10\,\text{s}$; panel c: average gap size is $\mu_G = 30\,\text{s}$.





In order to re-interpret the GP-based linear interpolation uncertainty formula of Figures 3 and 4, we consider the ensemble of all the estimated local GP model parameters set $\mathbf{\Psi}$ from the entire cross-validation exercise. Coherently with the known small intrinsic error declared by Vaisala, the top panel of Figure 9 shows very small values of $\sigma_\epsilon$. Moreover, from the second panel of the same figure, we see that the values $\sigma_s < 1$ are common and in particular $\sigma_s = 0.5$ used in Figures 3 and 4 is quite plausible. Eventually, the bottom panel of Figure 9 shows that the correlation range $\theta$ may be easily between one and 15 minutes.

## 8.1 Interpolation distance

In general, the connection between the uncertainty curves of Figures 3 and 4 and the cross-validation evidence is worth to be studied. Considering both the gap size and the distance from the observations at various altitudes gives rise to hard-to-manage curve plots and a multiplicity of results. For this reason, the subsequent analysis is based on the "interpolation distance" in sec, which is denoted by $d$ and is given by the geometric mean of the temporal distances of each missing data from the closest observations $y^-$ and $y^+$ in the notation of Section 3.

Figure 10 depicts the cross-validation RMSE of the linear interpolation as a function of this distance by altitude, namely

$$MSE(d|ALT) = avg(e^2|d, ALT) \tag{12}$$

where $avg(\cdot|ALT, d)$ is the average of all the cross-validation terms with $alt \in ALT$ and interpolation distance $d$. We note that, in order to have high sampling information for both low and high interpolation distances, the graph is obtained by merging the block-bootstrap simulations obtained for $\mu_G = 10$ and $30\,\mathrm{s}$. Moreover, the graph is limited to 70" because there is a reduction of cross-validated data, especially at high altitudes. Of course, using the same approach, longer interpolation distances may be easily explored.

In addition, Figure 11 depicts the corresponding graph for the linear interpolation quadratic average of $SE(t^*) = SE(t^*|s, l, b)$, given by Equation (3) at cross-validation time $t^*$, station $s$, launch $l$ and bootstrap replication $b$, namely

$$\overline{SE(d|ALT)} = \sqrt{avg(SE(t^*|s,l,b)^2|d, ALT)}. \tag{13}$$

The corresponding graph for the GP-SE of Equation (9) is not reported here because, not only the two interpolation methods are exchangeable, as noticed above, but also their SE's give very close results, with a mean absolute difference between the two smaller than 0.005 K. It may be noted that, although the above two graphs have a similar increasing behaviour, the SE systematically underestimates the interpolation uncertainty. This is due, primarily to the GP model approximation for the present case study and, secondarily, to estimation uncertainty. Hence, we propose a corrected uncertainty estimate given by

$$u(t^*)^2 = SE(t^*)^2 + (MSE(d|ALT) - \overline{SE(d|ALT)}^2). \tag{14}$$

This semi-parametric bootstrap uncertainty estimate extracts information both from the average cross-validation performance at a certain altitude and interpolation distance and from the single profile behaviour approximated by the GP process.

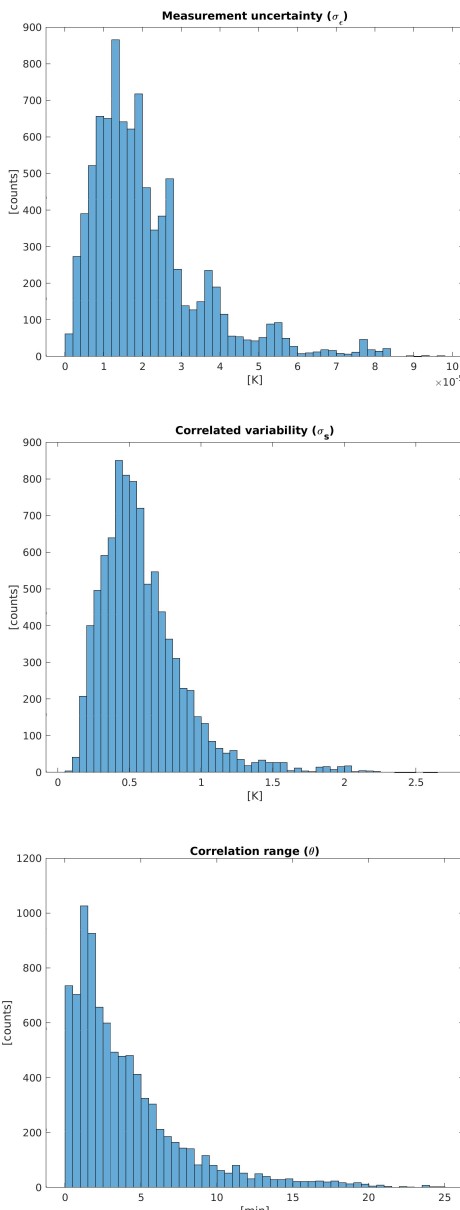

**Figure 9.** Distribution of estimated GP model parameters from all bootstrapped profiles and all atmospheric layers. Top panel: $\sigma_\epsilon$ [K]; centre panel: $\sigma_s$ [K]; bottom panel: correlation range $\theta$ [min]. The average gap size is $\mu_G = 10\,\mathrm{s}$ and the missing fraction is $f = 0.13$.



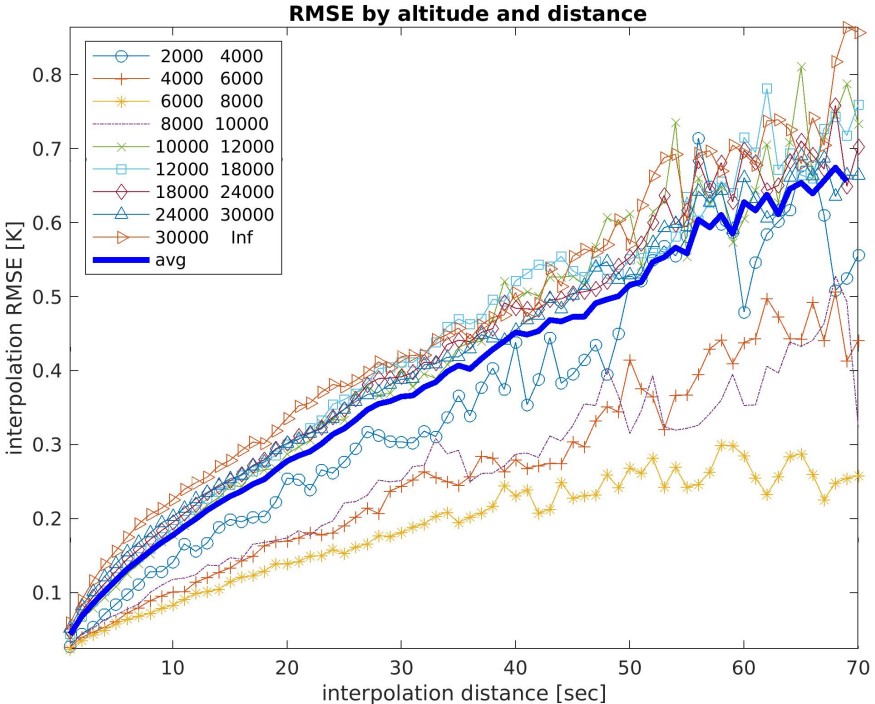

**Figure 10.** Cross-validation RMSE of linear interpolation by altitude [m] and interpolation distance [sec]. The $x$-axis is given by the geometric mean of the distances of each missing data from the closest "good" data, $y^-$ and $y^+$. The graph is obtained my merging the data sets with average gap sizes $\mu_G = 10$ and $30\,\mathrm{s}$.

## 8.2 Practical aspects

As an illustration of the method, the profile of Sodankylä site on 2017-03-03 12:00 is considered in Figure 12, left panel. This profile has T=4722 measurements and no original missing values. Using the block-bootstrap, 563 measurements have been
deleted and considered as pseudo-missing generating gaps between 1 and 24 s to be interpolated. From a practical point of view, such a missing rate and gap lengths can be considered a relatively common case, yet serious, for interpolation. Figure 12, right panel, shows both $\pm$ the GP uncertainty (7), and $\pm$ the Bootstrap uncertainty (14), computed at the interpolated pseudo-missing values. In doing this computation, formulas (12) and (13) are implemented as lookup tables (LUT) with entries geometric distance and altitude. Figure 13 focuses on the above profile around 22 Km height and shows the interpolation uncertainty of
a single point gap, two small gaps and three larger gaps.

It follows that the implementation of a GRUAN data processing giving interpolated temperature profiles with their uncertainty requires some efforts which are divided into two different phases. First, a massive GP off-line computation is needed



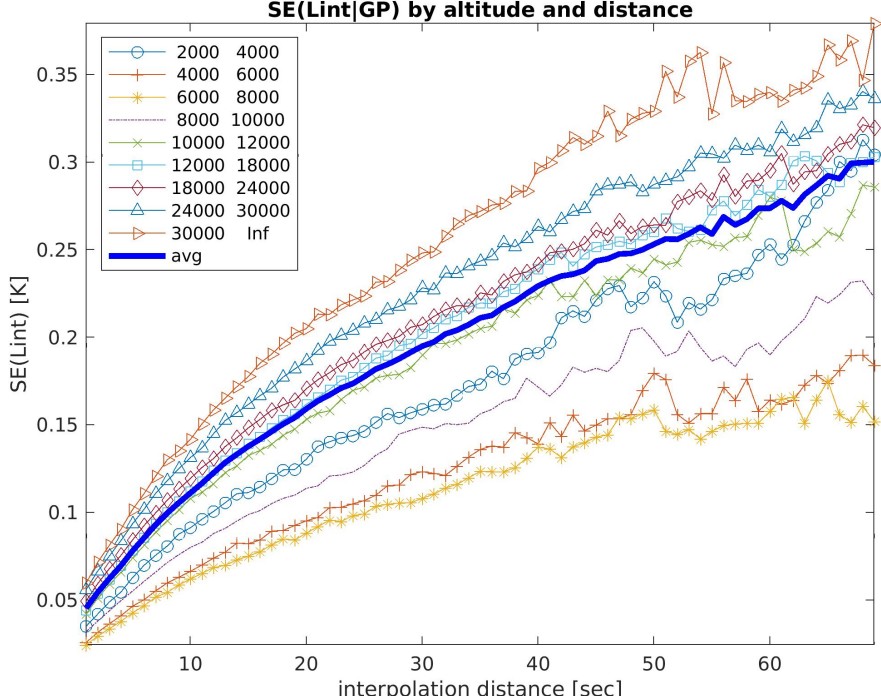

**Figure 11.** Linear interpolation SE by altitude [m] and interpolation distance [s]. The depicted SE is the quadratic average of formula (7) for each altitude and interpolation distance in the validation data set. The interpolation distance is given by the geometric mean of the distances of each missing data from the closest "good" data, $y^-$ and $y^+$. The graph is obtained my merging the data sets with average gap sizes $\mu_G = 10$ and $30\,\mathrm{s}$.

to prepare the LUT related to Equations (12) and (13). Second, for each profile an on-line local GP calibration is needed to provide the SE (7) for the interpolated values. After that, Equation (14) easily gives the corrected interpolation uncertainty.

## 9  Conclusions

This paper gives a multifaceted assessment of the interpolation uncertainty of Vaisala RS41 temperature profiles at various altitudes using an extensive data set coming from seven GRUAN stations. Moreover, it provides a general frame for interpolation of generic atmospheric profiles. Two complementary approaches are developed and integrated.

The first one is a cross-validation approach based on block-bootstrap, which shows that the average of the root mean square error of linear interpolation is about 0.1 K for small gaps and increases up to 0.58 K for gaps of an average duration of 60". These results may be made operational as lookup tables characterising interpolation uncertainty with entries altitude and 'interpolation distance'.

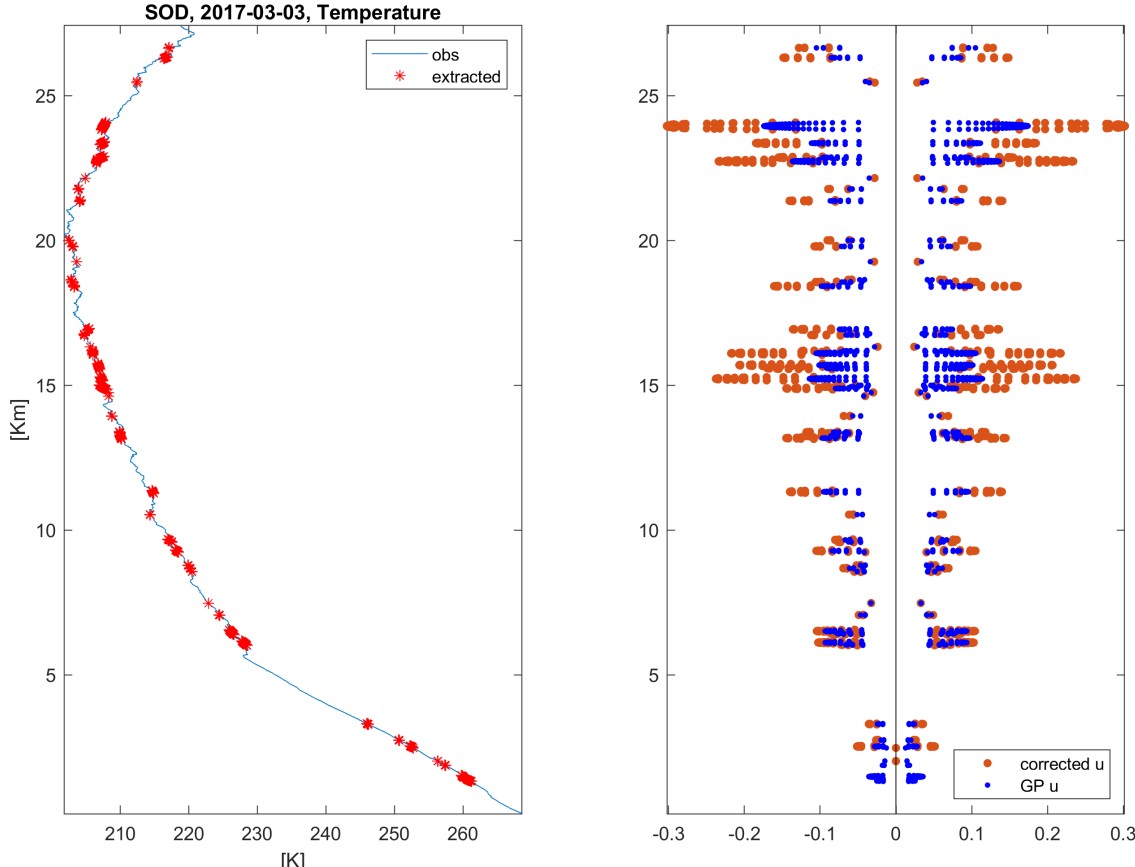

**Figure 12.** RS42 temperature profile at Sodankylä site on 2017-03-03 12:00. Left panel: observation is in blue and block-Bootstrap pseudo-missing are the red stars. Right panel: $\pm$ linear interpolation uncertainty of pseudo-missing values; GP uncertainty (7) is in blue; Bootstrap uncertainty (14) is in orange.



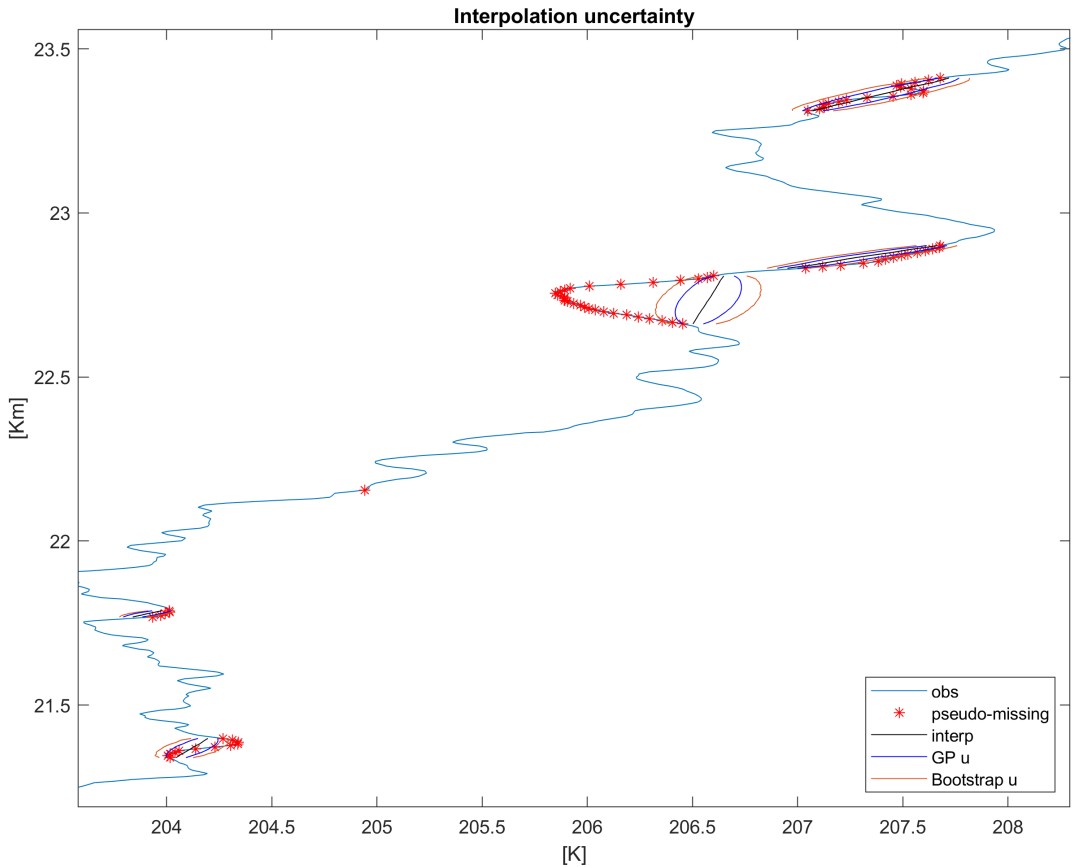

**Figure 13.** Detail of RS42 temperature profile at Sodankylä site on 2017-03-03 12:00, around 23 Km altitude. Observation is in blue and block-Bootstrap pseudo-missing are the red stars; linear interpolation is in black; $\pm$ GP uncertainty (7) is in blue; $\pm$ Bootstrap corrected uncertainty (14) is in orange.





Since the cross-validation outputs are averages, the individual contribution to the uncertainty is not considered. Hence, the second approach addresses this point using Gaussian Process computations. This allows obtaining two formulas for the interpolation uncertainty. One is the uncertainty of the linear interpolation, and the other one is based on GP-interpolation. For the Vaisala RS41 high-vertical-resolution data, they result to be very close and, hence, interchangeable.

Integrating the above two approaches, a bootstrap-corrected formula for the individual interpolation uncertainty is proposed. Based on these results, GRUAN data processing could implement interpolated temperature profiles, uncertainty included.

The extension of this approach to other essential climate variables (ECV) and/or other instruments requires some considerations. From the modelling point of view, provided enough field data are available, the extension is relatively straightforward. In fact, the approach is quite general, and model selection and optimisation are data-driven. Hence similar results may be expected for temperature profiles obtained by other instruments, provided that vertical resolution and instrumental error are comparable to the present case. Further, similar results are also expected for other smooth variables, such as pressure.

On the other side, the interpolation uncertainty could be larger for those ECV which are known to have large variations also in the small scale. For example, relative humidity commonly shows highly intermittent profiles in the troposphere with very large and very fast changing gradients. In these cases, we can expect that the cross-validation uncertainty could be large even for small gaps. In addition, the vertical autocorrelation could have a shorter range and the corresponding GP model could provide interpolation uncertainties close to the white noise case considered in Section 3.





*Code and data availability.*  TEXT

The underlying MATLAB code is available from the author upon request. The data are available from the GRUAN Lead
Center, www.gruan.org.

## Appendix A:  Linear interpolation uncertainty

To see Equation (7), let us rewrite the interpolation error of Equation (6) as follows:

$$m(t) - s(t) = \alpha y^+ + (1-\alpha)y^- - (y(t) + \epsilon(t)) = \boldsymbol{a}'\boldsymbol{u}$$

where $\alpha(t) = \frac{t-t^-}{t^+-t^-}$ as in Section 3.1, $\boldsymbol{a}' = (\alpha(t), 1-\alpha(t), -1, +1)$ is a vector of constants for fixed times $t^- \leq t \leq t^+$ and
$\boldsymbol{u}' = (y(t^+), y(t^-), -y(t), +\epsilon(t))$ is a stochastic vector. With these symbols, Equation (7) may be written as:

$$SE(t)^2 = E(m(t) - s(t))^2 = \boldsymbol{a}'\boldsymbol{\Sigma_u}\boldsymbol{a}$$

where $\boldsymbol{\Sigma_u}$ is the variance-covariance matrix of $\boldsymbol{u}$ given by

$$\boldsymbol{\Sigma_u} = \begin{bmatrix} \sigma_y^2 & \gamma(t^+ - t^-) & \gamma(t^+ - t) & 0 \\ & \sigma_y^2 & \gamma(t^- - t) & 0 \\ & & \sigma_y^2 & 0 \\ & & & \sigma_\epsilon^2 \end{bmatrix}.$$

The conclusion follows by straightforward algebra.

*Competing interests.*  The authors declare that they have no conflict of interests

*Acknowledgements.*  The authors wish to thank the GRUAN QTF group for the extensive discussions.





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
