# Peer review of "Interpolation uncertainty of atmospheric temperature profiles"

_Atmospheric Measurement Techniques, 2020_

## Referee Comment (RC1) · Anonymous Referee #1 · 28 Jul 2020

General comments:

This paper addresses an important question that researchers come across when aiming to estimate the uncertainty on interpolated profile data. The paper is a valuable contribution to Atmospheric Measurements Techniques and I would recommend the publication of this manuscript after a minor revision, which should include some clarifications and improvements to the figures. It is clear that much effort has been put into this work and I' m sure it will be a valuable contribution to the atmospheric research community.

In the following, I will answer the questions posed on the AMT website to be answered in the review and then provide specific comments and technical corrections. 1. Does the paper address relevant scientific questions within the scope of AMT? Yes.

[Figure]

2. Does the paper present novel concepts, ideas, tools, or data? Yes.

3. Are substantial conclusions reached? Yes.

4. Are the scientific methods and assumptions valid and clearly outlined? I am not a statistician and may therefore not be qualified if all assumptions and equations are valid. However, to the best of my knowledge, the methods seem to be valid. I would describe the explanation suitable for an expert user and could mostly follow the steps.

5. Are the results sufficient to support the interpretations and conclusions? Yes.

6. Is the description of experiments and calculations sufficiently complete and precise to allow their reproduction by fellow scientists (traceability of results)? Without trying this myself, I don't feel in the position to confidently answer this question.

7. Do the authors give proper credit to related work and clearly indicate their own new/original contribution? Yes.

8. Does the title clearly reflect the contents of the paper? Yes, however, I will suggest an improvement below.

9. Does the abstract provide a concise and complete summary? Yes.

10. Is the overall presentation well structured and clear? Yes.

11. Is the language fluent and precise? Yes, mainly. I have suggested some improvements below.

12. Are mathematical formulae, symbols, abbreviations, and units correctly defined and used? I suggested some changes to the units and abbreviations that have not been introduced, but overall the standard is good.

13. Should any parts of the paper (text, formulae, figures, tables) be clarified, reduced, combined, or eliminated? Tables and Figures: Your table and figure captions are rather scarce and need to be improved to ensure the reader does not need to guess what has

been shown. I made some specific comments for individual plots, but please also have another careful look over them. I also suggested in several instances to use common axis limits to aid the reader in understanding the plots.

14. Are the number and quality of references appropriate? Yes.

15. Is the amount and quality of supplementary material appropriate? Yes.

Specific comments:

Title: I wonder if the title could be improved, e.g. 'Interpolation uncertainty of atmospheric temperature profiles' or 'Interpolation uncertainty of atmospheric temperature profiles measured with radiosondes'. The reason for this suggestion is, that I would not search for the word 'radiosoundings' but rather for radiosonde or for profile.

Line 18: I would argue it is not just the ULTS, but the entire troposphere and stratosphere

Line 22: Also, here I would argue that GRUAN provides reference observations from the surface, through the troposphere, and into the stratosphere (rather than in the UTLS)

Line 30: Please rewrite sentence starting on this line e.g. "If interpolation is applied to fill in the missing values, the uncertainty introduced through interpolation should be taken in the uncertainty budget."

Line 44: RAOB is not a network, but a software/program, see https://www.raob.com/. I would take out any mentioning of RAOB as I do not think this is useful. Some people use RAOB as an abbreviation for radiosonde observations, however, RS may be better suitable for your paper as you are using the RS41 radiosonde. What you probably mean is the operational radiosonde network. However, nowadays most radiosondes measure with a high vertical resolution, so it is more for historical reasons as the old TEMP format is providing low resolution profiles. An increasing number of sites now report data in high resolution BUFR format.

[Figure]

Line 57: The sentence starting on this line is not clear to me. Either take it out or try to improve please.

Line 71: ' the latter being used as missing data for interpolation uncertainty assessment' if I understand it right, you are not using the testing set to assess the interpolation uncertainty, but rather the lack of this data is used to assess the uncertainty. Thus, please reword.

Line 73: Could you please give a reference for 'block-bootstrap cross-validation scheme'?

Line 90: I think a better word for 'station' in this context is 'site' which is also used on the GRUAN website. I would suggest correcting that throughout the paper.

Line 117: Please explain what e(t) is.

Line 118: I don't understand what you mean with a local dynamics. Could you please clarify?

Line 141: Is E the expectation value? Please clarify.

Line around 203: Atmospheric output layering. Could you please clarify what you mean?

Line 216: I do not really understand this sentence. Please clarify what do you mean with the layering problem? Have you then selected a 400 second layer? If so, please write second out. However, please rewrite the entire sentence as I also am unsure what you mean with the atmospheric adaption. You may need to explain more what you mean so that the reader can follow you.

Line 231: Please clarify what you mean with 'overall linear interpolation uncertainty'? Is that an average of the two approaches or which approach is shown here?

Line 236: I would suggest to always use 's' as unit for second and not " in some instances. s is the SI unit for second and is therefore preferable and clearer. Most

[Figure]

importantly though, be consistent throughout the entire paper including the Figures.

Figure 10: As mentioned for other Figures as well, your figure caption should be improved. It should be clarified that the different lines show ranges of altitudes. Please use [s] as the unit of second based on SI standard as done earlier in the paper. Either in the caption, or in the text, can you please explain how interpolation distances of 70s are possible. Does that happen by merging the 30 and 10 second gap sizes? Delete the title of the Figure, as the description should be given in the caption.

Figure 11: As you are comparing this Figure to Figure 10, it would be helpful to have the same y-axis limits. You use "Lint" in the Figure labels, but have never introduces this abbreviation, please make sure you always introduce them. Also, please include the suggestions made at Figure 10. Delete the title of the Figure, as the description should be given in the caption.

Line 270: You state that the SE systematically underestimates the interpolation uncertainty, however, looking at the plots, they initially look very similar. Then I noticed that you did not use the same scale, which does make it harder to understand. Please correct this for the readers ease.

Line 270: Please also explain why you expect the standard error SE and the RMSE to be about equal? This is probably obvious to you but may not be to the reader and therefore a sentence to refresh that in peoples thinking would help.

Line 282: What I am missing here is a discussion of the results of the Figure 13. Depending on where in the profile values are missing, the estimated uncertainties are able to enclose the pseudo-missing data or not. At the fourth gap from the bottom, even the corrected, bootstrap does not agree with the actual measurement, as the measurements were missing in a local minimum of the profile. This should at least be discussed as this is a fundamental issue with interpolation which cannot be resolved easily.

Line 299 and 300: It again confuses me that you are first stating the two approaches are interchangeable and then, in the next line, you explain that you suggest a bootstrap corrected formula to integrate both approaches. There is something missing to clarify this.

Line 311: Is there any way the proposed method can be used by individual researchers? I.e. when using radiosonde data in the old, low-resolution TEMP format and interpolating to a given pressure level. Could lookup tables produced by GRUAN help to estimate this uncertainty as well? Or could lookup tables for averaged interpolation uncertainties be provided to give an estimate to other researchers?

Technical corrections:

Line 12: 'Since both . . .' Please clarify that you mean both approaches here.

Line 27: high vertical resolution

Line 38: I think "contemplate" may not be the right word here as it is another word for "consider" indicating it is not applied in OSSSMOSE.

Line 48/49: "in this frame" maybe better say: in this publication

Line 53: use 'similar' instead of 'the same' and delete 'to a large extend'

Line 61: correct 'note' to 'noted'

Line 63: replace 'integrating' with 'using'

Line 76: Correct spelling of 'soruces' to 'sources'

Line 95: correct 'radiosonde's' to 'radiosonde'

Line 96: Write out GNSS

Line 96: Maybe better: The raw data are corrected for known or experimentally evidenced systematic effects such as adjustments from . . .

Line 104: Please try to reword this sentence

Line 139: The formatting here is weird. Is the equation meant to be in-line? Or is it lacking a number? This seems to be a recurring problem in the following equations, i.e. some have an own line, but no number. Something also went wrong with the line numbering between 140 and 145 and at later instances where equations are present. In those instances, I provide approximate line numbers.

About line 144: I don't understand the sentence starting with: Since, using field... Please rewrite

Line 149: Try to reword 'with abuse of'

Line 162: write Equation (1) instead of model (1)

Line 182: replace 'little missing gaps' with 'few missing data'

Table 1: I would suggest changing the table caption to something along the lines of: GRUAN sites included into the Few_nan dataset and the respective number of launches (imported) and the number of profiles that have less than....

Table 1: I can only guess from the text and comparing the numbers what the column 'Imported' and 'selected' means, please clarify. Also, neither Ny-Ålesund nor Payerne are spelled as on the GRUAN website.

Line around 187: Please correct the sentence starting with: Mudelsee (2014) . . . as it misses a verb.

Line three below 187: take out 'able'

Line 216: Use were instead of where in: . . .the results were. . .

Line 218 . . . RMSE compared to. . .

Line 223: Maybe better: . . .set-up, we decided to use or settled on using the simplest . . .

Line 225: champaign – I assume you picked the wrong word here.

Line 238: be more specific in: . . . have slightly larger values. . . about what you mean with values, i.e. it is the RMSE.

Table 2: This table should come before Figure 7.

Line 242: Could you please state for which interpolation method you will show the plots in the latter?

Line 245: Maybe better . . . Lauder typically have the largest values.

Figure 8: You use the abbreviation CV in the figures but have not introduced it. Also, in the label you say that the x-axis is based on the RMSE. If I understand correctly what you did, it us the RMSE in [K]. Please reword. To clarify the effect, I suggest fixing the x-axis limits, or at least mention in the caption that the limits change from one panel to the next.

Line 261: Please write 70 seconds (or s) rather than 70".

Line 267: Please rewrite the long sentence starting on this line.

Line 269: "above two graphs" please give the Figure numbers instead as they may have moved around during the process and I assume you mean Figure 10 and Figure 11 which are below.

Line 278: Maybe better '. . .deleted (pseudo-missing)' as this sentence is currently not nice.

Line 280: Write (Equation 7) and (Equation 15)

Line 282: The SI unit for kilometre is km rather than Km.

Line 282: Here you state the plots shows values around 22km and later you state around 23km. While both is not wrong, it would be nice to be consistent.

Line 294: Write 60s or 60 seconds instead of 60"

In Acknowledgements: Please say what the QTF group is as this abbreviation has not been introduced.

Thank you very much for this nice research article!
* * *

---

## Referee Comment (RC2) · Anonymous Referee #2 · 30 Jul 2020

GENERAL COMMENTS

This manuscript contributes to highlight the importance of properly assessing uncertainty when imputation of missing data in atmospheric profiles is realized by interpolation, with novel ideas and tools within the scope of AMT. The statistical approach adopted is innovative, very valid and multifaceted, and it leads to reach substantial conclusions, even illustrated in its practical aspects. The overall presentation is well structured and clear, although some points need further clarifications and improvements, as specified in the following. Thus I would recommend the publication of this manuscript after a minor revision, believing it will be very interesting and useful for AMT readers.

SPECIFIC COMMENTS

[Figure]

Figures: titles are redundant since information is already written in the caption, if necessary please add information in the caption but remove titles; moreover check axis names (e.g. missing in Figures 2-4) and limits

Line 65: please clarify the sentence "thanks to the availability of appropriate data"

Lines 118-120: the error term epsilon(t) should be introduced after its appearance in Equation (1)

Lines 118- 124: although the assumption of GP is relaxed in a second phase, it would be suitable to justify or at least comment upon the choice of the two considered auto-covariance functions

Line 127: here the assumption of zero expected value for the error term in Equation (1) is implied, while it could be written before

Line 174: the assumption of a GP as a good description of the problem comes with a specified autocovariance function, it would be useful to clarify this

Line 200: to facilitate reading, it would be useful to specify that $m_1$ refers to Equation (5) and $m_2$ to Eq. (9), and only for $m_2$ we need an estimation method (and so a hat)

Section 8: this section needs to be revised because a 3x3 simulation design is described but after there are comments about the 60 seconds case (e.g. line 237) and even results (e.g. in Table 2). Please correct consistently to have a 4x3 or 3x3 simulation design in all section

Line 228: I would suggest to avoid the technical term "2-fold" since it is not introduced before and not necessary

Line 270: it could be useful to clarify this sentence, consistently with the abstract where you state that both interpolation methods provide an underestimation

Lines 299-300: it could be useful a line summarizing reasons to integrate the two approaches and then use the proposed bootstrap-corrected formula

TECHNICAL CORRECTIONS

Line 44: a reference for RAOB would be useful

Line 61: "note" should be "noted"

Line 76: "soruces" should be "sources"

Line 89: a reference for the statistical analysis conducted by GRUAN would be useful

Table 1: please specify what "Imported" and "Selected" mean

Line 186: add parenthesis for the two references

Line 225: maybe campaign?

Line 237: "vey" should be "very"

Figure 7: in addition to the general comments about figures, this one appears with a different look, I would suggest to use the same software for all plots

Line 261: in the same line seconds are written differently, please check throughout all manuscript

Figures 10 and 11: captions should be revised since altitude is not represented as axis, and the box with altitude intervals needs a title

Line 282 and Figure 13: please write in both points 22 or 23 km

Figure 11: "Lint" needs to be defined

Figure 12: axis names are missing

Figure 13: please change one blue color

Line 312: please delete "TEXT"

Line 325: please explain QTF

Line 335: Finazzi et al. should have year 2019 (that is correct at page 2)

---

## Author Comment (AC1) · 23 Sep 2020

"Interpolation uncertainty of atmospheric temperature radiosoundings"

Answer to Referee #1

Dear Referee,

Thanks a lot for using your precious time in reading our paper and making valuable comments that, we believe, are relevantly improving the paper.

Generally speaking, we appreciated all your comments and changed the manuscript accordingly. Also, note that the professional proofreading of the article has been made.

In the sequel, we answer point by point to your report. All answers below and new text parts in the manuscript are in blue, except changes related to the 2nd referee, which are in green.

Best regards, Alessandro Fassò

**General comments:**

This paper addresses an important question that researchers come across when aiming to estimate the uncertainty on interpolated profile data. The paper is a valuable contribution to Atmospheric Measurements Techniques and I would recommend the publication of this manuscript after a minor revision, which should include some clarifications and improvements to the figures. It is clear that much effort has been put into this work and I'm sure it will be a valuable contribution to the atmospheric research community.

ANSWER: ... thanks ... overall changes ... captions and titles of figures and tables have been reworked ...

**Specific comments:**

Title: I wonder if the title could be improved, e.g. 'Interpolation uncertainty of atmospheric temperature profiles' or 'Interpolation uncertainty of atmospheric temperature profiles measured with radiosondes'. The reason for this suggestion is, that I would not search for the word 'radiosoundings' but rather for radiosonde or for profile.

ANSWER: Thanks for this suggestion, we adopted the title 'Interpolation uncertainty of atmospheric temperature profiles'.

Line 18: I would argue it is not just the ULTS, but the entire troposphere and stratosphere

ANSWER: Thanks, we removed UTLS. Now, we start from the more generic "atmosphere" and, then, we make it more precise in the next point.

Line 22: Also, here I would argue that GRUAN provides reference observations from the surface, through the troposphere, and into the stratosphere (rather than in the UTLS)

ANSWER: we adjusted accordingly.

Line 30: Please rewrite sentence starting on this line e.g. "If interpolation is applied to fill in the missing values, the uncertainty introduced through interpolation should be taken in the uncertainty budget."

ANSWER: Thanks for this. It is more clear and linear.

Line 44: RAOB is not a network, but a software/program, see https://www.raob.com/. I would take out any mentioning of RAOB as I do not think this is useful. Some people use RAOB as an abbreviation for radiosonde observations, however, RS may be better suitable for your paper as you are using the RS41 radiosonde. What you probably mean is the operational radiosonde network. However, nowadays most radiosondes measure with a high vertical resolution, so it is more for historical reasons as the old TEMP format is providing low resolution profiles. An increasing number of sites now report data in high resolution BUFR format.

ANSWER: We removed references to RAOB, and clarified that this is relevant for historical data sets.

Line 57: The sentence starting on this line is not clear to me. Either take it out or try to improve please.

ANSWER: We removed the sentence. In fact, commenting on the relationship between interpolation and forecast is not necessary for this paper.

Line 71: ' the latter being used as missing data for interpolation uncertainty assessment' if I understand it right, you are not using the testing set to assess the interpolation uncertainty, but rather the lack of this data is used to assess the uncertainty. Thus, please reword.

ANSWER: We reworded the paragraph to make it more clear.

Line 73: Could you please give a reference for 'block-bootstrap cross-validation scheme'?

ANSWER: The concept of 'block-bootstrap cross-validation scheme' is discussed in Section 5, so we removed this sentence.

Line 90: I think a better word for 'station' in this context is 'site' which is also used on the GRUAN website. I would suggest correcting that throughout the paper. ANSWER: Done.

Line 117: Please explain what e(t) is.

ANSWER:  $\epsilon(t)$  is the white-noise measurement error. We rephrased and explained it.

Line 118: I don't understand what you mean with a local dynamics. Could you please clarify?

ANSWER: In the 1st manuscript, "locally" referred to the atmospheric conditions and "dynamics" to the autocorrelation. In the revised one, this sentence is rephrased more explicitly.

Line 141: Is E the expectation value? Please clarify.

ANSWER: Yes it is. Now, the interpolation uncertainty formula is introduced with a more clear text.

Line around 203: Atmospheric output layering. Could you please clarify what you mean?

ANSWER: Each single measurement is taken at certain altitude, say *alt*, known with good precision. The same holds for the errors in line 227 of 1st manuscript. *ALT* identifies a layer of the atmosphere with 1 km thickness. Hence for a given layer *ALT*, the MSE in (11) is given by the average of the squared errors of this layer.

The term "output" is used to differentiate this layering wrt the model segmentation, you noticed in the next comment.

The text has been improved accordingly.

Line 216: I do not really understand this sentence. Please clarify what do you mean with the layering problem? Have you then selected a 400 second layer? If so, please write second out. However, please rewrite the entire sentence as I also am unsure what you mean with the atmospheric adaption. You may need to explain more what you mean so that the reader can follow you.

ANSWER: Thanks for pointing this lack of clarity. We substituted "the layering problem" with "the choice of the layering resolution".

Line 231: Please clarify what you mean with 'overall linear interpolation uncertainty'? Is that an average of the two approaches or which approach is shown here? ANSWER: We removed the confusing term "overall" and added the reference to Eq. (11).

Line 236: I would suggest to always use 's' as unit for second and not " in some instances. s is the SI unit for second and is therefore preferable and clearer. Most importantly though, be consistent throughout the entire paper including the Figures. ANSWER: Thanks, we made it uniform all around the paper.

Figure 10: As mentioned for other Figures as well, your figure caption should be improved. It should be clarified that the different lines show ranges of altitudes. ANSWER: The caption has been improved.

Please use [s] as the unit of second based on SI standard as done earlier in the paper.

**ANSWER: Done.**

Either in the caption, or in the text, can you please explain how interpolation distances of 70s are possible. Does that happen by merging the 30 and 10 second gap sizes?

ANSWER: Distances of 70s are possible for an average gap size of  $\mu_G = 30$ s because the block-bootstrap generates random gaps with Geometric distribution. This is now briefly mentioned in the text after Equation (12).

Delete the title of the Figure, as the description should be given in the caption. ANSWER: Done.

Figure 11: As you are comparing this Figure to Figure 10, it would be helpful to have the same y-axis limits.

ANSWER: Done.

You use "Lint" in the Figure labels, but have never introduces this abbreviation, please make sure you always introduce them.

ANSWER: This acronym has been eliminated.

Also, please include the suggestions made at Figure 10. Delete the title of the Figure, as the description should be given in the caption. ANSWER: Done.

Line 270: You state that the SE systematically underestimates the interpolation uncertainty, however, looking at the plots, they initially look very similar. Then I noticed that you did not use the same scale, which does make it harder to understand. Please correct this for the readers ease.

ANSWER: Now, we make the two figures more comparable by using the same y-axis range. Moreover, it is remarked that SE > RMSE only at distance=1s, where both are very close to 0.

Line 270: Please also explain why you expect the standard error SE and the RMSE to be about equal? This is probably obvious to you but may not be to the reader and therefore a sentence to refresh that in peoples thinking would help.

ANSWER: If 1) the GP used was a "perfect" model for our data, 2) its coefficients  $\Psi$  where known and 3) the cross-validation was exact giving  $MSE_y(t)^2 = E[(m(t) - y(t))^2]$ , than, we would have:

$$u^2 = SE^2 = MSE - \sigma_\epsilon^2$$

Now,  $\sigma_{\epsilon}^2$  is very small, so we should have

 $SE^2 \approx MSE$

. The new version addresses this.

Line 282: What I am missing here is a discussion of the results of the Figure 13. Depending on where in the profile values are missing, the estimated uncertainties are able to enclose the pseudo-missing data or not. At the fourth gap from the bottom, even the corrected, bootstrap does not agree with the actual measurement, as the measurements were missing in a local minimum of the profile. This should at least be discussed as this is a fundamental issue with interpolation which cannot be resolved easily.

ANSWER: Thanks for pointing this. A discussion has be added, showing that even in the 4th case (D) the error is close to  $3 \times$  the bootstrap corrected uncertainty.

Line 299 and 300: It again confuses me that you are first stating the two approaches are interchangeable and then, in the next line, you explain that you suggest a bootstrap corrected formula to integrate both approaches. There is something missing to clarify this.

ANSWER: There are two alternative interpolation techniques, namely linear interpolation and GP-interpolation, which resulted to be equivalent for these data. Moreover there two complementary uncertainty approaches, the bootstrap based and the GP based, which have been integrated.

To avoid confusion, from the conclusions we removed the distinction between linearand GP-interpolation.

Line 311: Is there any way the proposed method can be used by individual researchers? I.e. when using radiosonde data in the old, low-resolution TEMP format and interpolating to a given pressure level. Could lookup tables produced by GRUAN help to estimate this uncertainty as well? Or could lookup tables for averaged interpolation uncertainties be provided to give an estimate to other researchers?

ANSWER: This is an important point, which we are ready to answer only in a informal way. A lookup table published by GRUAN could be used for the average part of the interpolation uncertainty. A software could be developed for the GP computation.

However, the present study covered gap sizes in a range considered important for GRUAN processing (one minute gaps). This is different for example if we want to interpolate mandatory levels (several minute gaps).

**Technical corrections:**

Line 12: 'Since both ...' Please clarify that you mean both approaches here. Done. Line 27: high vertical resolution. Done.

Line 38: I think "contemplate" may not be the right word here as it is another word for "consider" indicating it is not applied in OSSSMOSE. Changed to "use".

Line 48/49: "in this frame" maybe better say: in this publication. We rephrased as follows: "These authors used spline interpolation of radiosonde profiles, and indirectly assessed the related uncertainty through a comparison with GRUAN reference measurements."

Line 53: use 'similar' instead of 'the same' and delete 'to a large extend'. Done. Line 61: correct 'note' to 'noted'. Done. Line 63: replace 'integrating' with 'using'. Done.

Line 76: Correct spelling of 'soruces' to 'sources'. Done.

Line 95: correct 'radiosonde's' to 'radiosonde'. Done.

Line 96: Write out GNSS. Done.

Line 96: Maybe better: The raw data are corrected for known or experimentally evidenced systematic effects such as adjustments from. Done.

Line 104: Please try to reword this sentence. Done.

Line 139: The formatting here is weird. Is the equation meant to be in-line? Or is it lacking a number? Done.

This seems to be a recurring problem in the following equations, i.e. some have an own line, but no number.

When appropriate for readability, we use displayed equations. On the other side, we number only those displayed equations which are referred in the text.

Something also went wrong with the line numbering between 140 and 145 and at later instances where equations are present. In those instances, I provide approximate line numbers. Thanks.

About line 144: I don't understand the sentence starting with: Since, using field. Please rewrite. It has been rephrased.

Line 149: Try to reword 'with abuse of'. This is a standard, see https://en. wikipedia.org/wiki/Abuse\_of\_notation.

Line 162: write Equation (1) instead of model (1). Done.

Line 182: replace 'little missing gaps' with 'few missing data'. Done.

Table 1: I would suggest changing the table caption to something along the lines of: GRUAN sites included into the Few\_nan dataset and the respective number of launches (imported) and the number of profiles that have less than. Done.

Table 1: I can only guess from the text and comparing the numbers what the column 'Imported' and 'selected' means, please clarify. Also, neither Ny-Ålesund nor Payerne are spelled as on the GRUAN website. Done.

Line around 187: Please correct the sentence starting with: Mudelsee (2014) as it misses a verb. Done.

Line three below 187: take out 'able'. Done.

Line 216: Use were instead of where in the results were. Done.

Line 218: RMSE compared to. Done.

Line 223: Maybe better set-up, we decided to use or settled on using the simplest. Done.

Line 225: champaign – I assume you picked the wrong word here. Oops ....

Line 238: be more specific in have slightly larger values about what you mean with values, i.e. it is the RMSE. Done.

Table 2: This table should come before Figure 7. Thanks, we adjusted the comment sequence accordingly.

Line 242: Could you please state for which interpolation method you will show the plots in the latter? Done.

Line 245: Maybe better ... Lauder typically have the largest values. Done.

Figure 8: You use the abbreviation CV in the figures but have not introduced it. Also, in the label you say that the x-axis is based on the RMSE. If I understand correctly what you did, it us the RMSE in [K]. Please reword. Thanks, we explained in the caption.

To clarify the effect, I suggest fixing the x-axis limits, or at least mention in the caption that the limits change from one panel to the next. Done.

Line 261: Please write 70 seconds (or s) rather than 70". Thanks, done.

Line 267: Please rewrite the long sentence starting on this line. Thanks, done.

Line 269: "above two graphs" please give the Figure numbers instead as they may have moved around during the process and I assume you mean Figure 10 and Figure 11 which are below. Thanks for pointing it, done.

Line 278: Maybe better '... deleted (pseudo-missing)' as this sentence is currently not nice. Thanks, we rephrased.

Line 280: Write (Equation 7) and (Equation 15). Done.

Line 282: The SI unit for kilometre is km rather than Km. Done.

Line 282: Here you state the plots shows values around 22km and later you state around 23km. While both is not wrong, it would be nice to be consistent. Done.

Line 294: Write 60s or 60 seconds instead of 60". Done.

In Acknowledgements: Please say what the QTF group is as this abbreviation has not been introduced. Done.

Thank you very much for this nice research article! We appreciated very much your careful reading and detailed commenting, and ... the "bon ton".

---

## Author Comment (AC2) · 23 Sep 2020

"Interpolation uncertainty of atmospheric temperature radiosoundings"

Answer to Referee #2

Dear Referee,

Thanks a lot for using your precious time in reading our paper and making valuable comments that, we believe, are relevantly improving the paper.

Generally speaking, we appreciated all your comments and changed the manuscript accordingly. Also, note that the professional proofreading of the article has been made.

In the sequel, we answer point by point to your report. All answers below and new text parts in the manuscript, specific to your comments only, are in green, while blue is for changes solicited by the 1st referee, or by you both, or (minor) changes independent on you.

Best regards, Alessandro Fassò

**General comments:**

This manuscript contributes to highlight the importance of properly assessing uncertainty when imputation of missing data in atmospheric profiles is realized by interpolation, with novel ideas and tools within the scope of AMT. The statistical approach adopted is innovative, very valid and multifaceted, and it leads to reach substantial conclusions, even illustrated in its practical aspects. The overall presentation is well structured and clear, although some points need further clarifications and improvements, as specified in the following. Thus I would recommend the publication of this manuscript after a minor revision, believing it will be very interesting and useful for AMT readers.

ANSWER: ... thanks ... overall changes ... captions and titles of figures and tables have been reworked ...

**Specific comments:**

Figures: titles are redundant since information is already written in the caption, if necessary please add information in the caption but remove titles; moreover check axis names (e.g. missing in Figures 2-4) and limits.

We eliminated all figure titles but the panel titles in multi-panel figures.

Line 65: please clarify the sentence "thanks to the availability of appropriate data".

Replaced with: thanks to the availability of "good" profiles without missing data. Lines 118-120: the error term epsilon(t) should be introduced after its appearance

in Equation (1).

Done in blue.

Lines 118- 124: although the assumption of GP is relaxed in a second phase, it would be suitable to justify or at least comment upon the choice of the two considered autocovariance functions.

Added some comments and a reference.

Line 127: here the assumption of zero expected value for the error term in Equation (1) is implied, while it could be written before.

The zero mean property has been stated earlier, close to Equation (1).

Line 174: the assumption of a GP as a good description of the problem comes with a specified autocovariance function, it would be useful to clarify this.

Added a reference to the role of the covariance.

Line 200: to facilitate reading, it would be useful to specify that  $m_1$  refers to Equation (5) and  $m_2$  to Eq. (9), and only for  $m_2$  we need an estimation method (and so a hat).

**Done.**

Section 8: this section needs to be revised because a 3x3 simulation design is described but after there are comments about the 60 seconds case (e.g. line 237) and even results (e.g. in Table 2). Please correct consistently to have a 4x3 or 3x3 simulation design in all section.

Done.

Line 228: I would suggest to avoid the technical term "2-fold" since it is not introduced before and not necessary.

Done.

Line 270: it could be useful to clarify this sentence, consistently with the abstract where you state that both interpolation methods provide an underestimation.

We reminded that the underestimation is for both the linear- and the GP-interploation. Lines 299-300: it could be useful a line summarizing reasons to integrate the two approaches and then use the proposed bootstrap-corrected formula.

We tried to make this point more clear by focusing mainly on the two uncertainties and blurring the diffrence among the linear- and the GP-interpolation, which resulted to be equivalent.

**Technical corrections:**

Line 44: a reference for RAOB would be useful. RAOB is discussed in Finazzi et al. (2019), already cited there.

Line 61: "note" should be "noted". Done.

Line 76: "soruces" should be "sources".Done.

Line 89: a reference for the statistical analysis conducted by GRUAN would be

useful. It is an internal preliminary study, we rephrased to make it more clear: "A preliminary statistical analysis for the occurrence of data gaps in RS41 radiosonde soundings performed at 15 GRUAN stations in the period 2014-2019 shows that gaps occur in more than 20% of the soundings, virtually independent of the height ranges, with the majority (> 95%) having less than 15 gaps per 1000 s."

Table 1: please specify what "Imported" and "Selected" mean. Done.

Line 186: add parenthesis for the two references. Done.

Line 225: maybe campaign? Done.

Line 237: "vey" should be "very". Done.

Figure 7: in addition to the general comments about figures, this one appears with a different look, I would suggest to use the same software for all plots. Done.

Line 261: in the same line seconds are written differently, please check throughout all manuscript. Done.

Figures 10 and 11: captions should be revised since altitude is not represented as axis, and the box with altitude intervals needs a title. Done.

Line 282 and Figure 13: please write in both points 22 or 23 km. Done.

Figure 11: "Lint" needs to be defined. Done.

Figure 12: axis names are missing. Done.

Figure 13: please change one blue color. Done.

Line 312: please delete "TEXT". Done.

Line 325: please explain QTF. Done.

Line 335: Finazzi et al. should have year 2019 (that is correct at page 2). Done.

---

## Referee Report (RR1)

Thank you very much for the careful reworking of your paper. I read your revised submission and you will find a list of my notes below, which should be easy to implement and enhance the consistency of your paper. I hope you will find these minor technical suggestions useful.

My technical recommendations:
You sometimes use a space between the number and its unit and sometimes not. I would suggest making that consistent.
Line 91: gaps > 10s
Figure 1: Something seems to have gone wrong with the units in Figure 1. While e.g. the count is unitless, the gap length should have a unit (i.e. s) as well as the cumulative axis, i.e. %.
Figure 6: You use [sec] as the unit, but used [s] in your other figures. I would suggest to consistently use [s].
Line 255: is used since.
Line 256: I think it should mean: .compared to the squared exponential.
Line 265: This section's
Table 2: In the header, please correct ´´ to s. Also, please clarify what the last row of the table is (I assume it is the average).
Line 282: It's not clear which table is meant with `latter table´
Line 291: The units are missing for the values of sigma.
Line 305: Spell ´geometric´ with a small letter.
Figure 8: Unit Km may better be spelled km, as this is the SI unit. Also, is f given in % as in Figure 7, or is it actually a fraction? A fraction of 0.13 would be 13%, which is quite different to 0.13% if you would have forgotten the unit of f. Please be consistent if you use the same name. Please also clarify that for Figure 9.
Line 339: As far as the gap size increases.
Line 341: The sentence starting in this line is not quite clear to me. Have you introduced what 3u means? It is also not quite clear to me how you see that the bootstrap approach provides a sensible uncertainty estimate. It's of course a lot better than the other estimation, but as you mentioned the uncertainty in this case is very hard to estimate.
Line 364: (ECVs)

---

## Author Response (AR2)

Dear AE and Reviewer,

I made all the suggested minor changes.
I thank the reviewer again for his/her precious time spent in
careful and patient reading.

Best regards,
Alessandro Fassò